# Equilibrium Optimization Algorithm with Deep Learning Enabled Prostate Cancer Detection on MRI Images

**DOI:** 10.3390/biomedicines11123200

**Published:** 2023-12-01

**Authors:** Eunmok Yang, K. Shankar, Sachin Kumar, Changho Seo, Inkyu Moon

**Affiliations:** 1Department of Financial Information Security, Kookmin University, Seoul 02707, Republic of Korea; emyang@kookmin.ac.kr; 2Department of Computer Science and Engineering, Saveetha School of Engineering, Saveetha Institute of Medical and Technical Sciences, Chennai 602105, India; drkshankar@ieee.org; 3Big Data and Machine Learning Lab, South Ural State University, Chelyabinsk 454080, Russia; 4College of IBS, National University of Science and Technology, MISiS, Moscow 119049, Russia; kumar.s@misis.ru; 5Department of Convergence Science, Kongju National University, Gongju-si 32588, Republic of Korea; 6Department of Robotics & Mechatronics Engineering, Daegu Gyeongbuk Institute of Science & Technology (DGIST), Daegu 42988, Republic of Korea

**Keywords:** cancer diagnosis, prostate cancer, magnetic resonance imaging, equilibrium optimizer, deep learning

## Abstract

The enlargement of the prostate gland in the reproductive system of males is considered a form of prostate cancer (PrC). The survival rate is considerably improved with earlier diagnosis of cancer; thus, timely intervention should be administered. In this study, a new automatic approach combining several deep learning (DL) techniques was introduced to detect PrC from MRI and ultrasound (US) images. Furthermore, the presented method describes why a certain decision was made given the input MRI or US images. Many pretrained custom-developed layers were added to the pretrained model and employed in the dataset. The study presents an Equilibrium Optimization Algorithm with Deep Learning-based Prostate Cancer Detection and Classification (EOADL-PCDC) technique on MRIs. The main goal of the EOADL-PCDC method lies in the detection and classification of PrC. To achieve this, the EOADL-PCDC technique applies image preprocessing to improve the image quality. In addition, the EOADL-PCDC technique follows the CapsNet (capsule network) model for the feature extraction model. The EOA is based on hyperparameter tuning used to increase the efficiency of CapsNet. The EOADL-PCDC algorithm makes use of the stacked bidirectional long short-term memory (SBiLSTM) model for prostate cancer classification. A comprehensive set of simulations of the EOADL-PCDC algorithm was tested on the benchmark MRI dataset. The experimental outcome revealed the superior performance of the EOADL-PCDC approach over existing methods in terms of different metrics.

## 1. Introduction

Prostate cancer (PrC) is the second most common cause of death among males and the most frequently diagnosed cancer in males around the world [1]. Earlier diagnosis of PrC is essential for more effective disease management. In several countries, the standard practice for PrC diagnosis relies on high rates of prostate-specific antigen (PSA) in the blood and a digital rectal examination (DRE). In some cases, pre-biopsy magnetic resonance imaging (MRI) may be recommended to guide the biopsy process [2]. Automated computer-aided diagnosis (CAD) and identification systems can address the limitations of the standard radiological analysis by applying quantitative techniques for automated, standardized, and supported regenerative analyses of radiological images [3]. Although PrC has a wide variety of cancers when diagnosed in the earlier phases, the survival rates are increased because of the gradual development of cancer [4]. As a result, efficient monitoring and earlier identification are significant for enhanced survival of patients.

Current research has found that implementing the machine learning (ML) technique in prostate MRI can enhance diagnosis accuracy and decrease inter-reader changeability by focusing on suspected regions on the MRI [5], permitting a more attentive analysis by the radiologist in standard scan analysis. The ML method is also capable of forecasting cancer aggressiveness and therapeutic reactions [6]. Numerous studies have indicated similar implementations among ML techniques and proficient radiologists in head-to-head relations for MRI analysis. The ML approach has a domain of artificial intelligence (AI) to employ statistical methods for learning hidden models from present data and decision making around hidden records [7]. The major function of a machine learner is to make an overall system on the probable allocation of training samples and, further, generalize capability to unseen samples. The learning method relies on the presented data quality. A sample exists in a database with various features. Effective feature extraction is unfortunately challenging for any task [8]. A convolutional neural network (CNN) method could have greater individual effectiveness in real image analyses, which are expected to improve CAD in prostate MRIs. The CNN-based DL technique restructures and revolutionizes the present analytic model [9]. The main and real-time modules of medical prostate MRI analyses are diffusion-weighted imaging (DWI), T2-weighted imaging (T2WI), and apparent diffusion coefficient (ADC) classifications. Numerous earlier studies of the DL method engaged a PC analysis employing only one or two of the aforementioned systems and, therefore, were not directly compared with medical efficiency [10].

The study proposes an Equilibrium Optimization Algorithm with Deep Learning-based Prostate Cancer Detection and Classification (EOADL-PCDC) technique applied to MRIs. The main goal of the EOADL-PCDC method lies in the recognition and categorization of prostate cancer. To achieve this, the EOADL-PCDC technique applies image preprocessing to improve the image quality. In addition, the EOADL-PCDC technique follows the CapsNet (capsule network) for the feature extraction approach. The EOA-based hyperparameter tuning increases the efficiency of the CapsNet model. For PrC classification, the EOADL-PCDC technique makes use of a stacked bidirectional long short-term memory (SBiLSTM) model. A comprehensive set of simulations of the EOADL-PCDC technique was tested on the benchmark MRI data.

## 2. Related Works

Singh et al. [11] implemented the DL technique for diagnosing PrC by employing the notion of Gleason grading. A 3D-CNN is utilized for monitoring the affected area and forecasting the cancerous area by employing an Epithelial and Gleason grading network. In [12], a hybrid method was developed for accurately analyzing mpMRI inspection and forecasting PI-RADS scores. In this developed technique, feature mapping of mpMRI is extracted with the help of Darknet53, MobilenetV2, and Efficientnetb0 frameworks. Further, the feature maps acquired utilizing the above-mentioned frameworks are incorporated. The combined feature maps have been subjected to neighborhood components analysis (NCA) for extracting redundant features. Li et al. [13] introduced an imaging pattern that depends on multi-parameter MRI. Primarily, a multi-view radiomics framework approach was developed. Secondarily, logistic regression algorithms were exploited for extracting features and making a framework. Lastly, a Swin Transformer model was developed and trained through transfer learning (TL) methods.

In [14], a technique for automated classification of the prostate area and cancerous area with the help of SegNet, a deep-CNN (DCNN) framework, was introduced. This method employs the PROSTATEx database to train the framework and incorporate various categorizations into three channels of one image. For all subjects, every part consisting of the PCa area, transition zone (TZ), and peripheral zone (PZ) is preferred. Ye et al. [15] presented a PrC identification that depends on the DL method, PSP-Net+VGG16.

The DCNN segmentation technique dependent upon the PSP-Net was used to make an atrous convolutional residual architecture framework extraction network. Primarily, a 3D prostate MRI was transformed into 2D image parts and trained to rely on the PSP-Net and the VGG_16 framework, which could be employed for detecting the targeted area and classifying the normal prostate and PrC. Ragab et al. [16] presented an Archimedes Optimizer Algorithm with DL-based PrC Classification (AOADLB-P2C) technique to execute preprocessing in two phases. Primarily, the technique extracts features through a DenseNet-161 architecture with the RMSProp optimization. Lastly, the method categorizes PCa utilizing the AOA with the least-square SVM (LS-SVM) approach.

In [17], a technique for computational identification of EPE on multivariant MRI exploiting DL was developed. This technique can include two phases. Initially, DL algorithms are trained by implementing the MRI as input for producing cancerous possibility maps both outside and inside the prostate. Next, this analysis makes an image postprocessing pipeline to produce estimates for the EPE position. Bouslimi and Echi [18] suggested a CAD technique. This developed method examines the convolutional neural networks (UNet) algorithm to identify PCa cancers and segmentation for accurately collecting cancer categories. This research provides a completely automated technique through MultiResUNet, which was primarily developed for the segmentation of skin cancers.

In [19], a new automated classification model was developed that combined various DL methods to identify PrC from ultrasound (US) and MRI images. Furthermore, the presented approach describes why a certain decision is made in the input of US or MRI images. In [20], a strong DL-CNN is applied using a transfer learning algorithm. The outcomes are compared to different ML approaches (different kernels, Decision Tree (DT), and SVM). In [21], a new DL model is used for creating a pipeline for the classification and segmentation of MRI images. The two steps of the DL technique are given as follows: a U-Net model to segment ROI in the first stage and an LSTM model for categorizing the ROI as non-cancerous or cancerous.

Several automated tools have been proposed in this work for efficient prostate cancer detection and classification. Although ML and DL techniques exist in the prior research, still it is essential to increase the efficiency of PrC classification. The number of parameters of DL techniques also quickly increases, leading to model over-fitting due to the continual deepening of the model. Simultaneously, dissimilar hyperparameters have a considerable influence on the CNN model performance. Thus, we apply EOA for the parameter selection of the CapsNet architecture. A summary of reviewed works is given in Table 1.

## 3. The Proposed Model

In this article, we introduce an automatic prostate cancer diagnosis model using the EOADL-PCDC technique on MRI Images. The main goal of the EOADL-PCDC algorithm lies in the detection and classification of PrC. The presented model involves four major stages, namely, image preprocessing, feature extractor, EOA-based hyperparameter tuning, and SBiLSTM-based classification. Figure 1 represents the overall working flow of the EOADL-PCDC technique.

### 3.1. Image Preprocessing

The CLAHE technique is used to improve the contrast level to preprocess the input images. It is an image enhancement method that has two fundamental aspects: (1) clip limit $\left(C_{limit}\right)$ and (2) non-overlapping regions $\left(Y_{contextual}\right)$ [22]. In this model, both parameters are accountable for controlling the improved quality of the image. $X_{av}$ refers to the average number of pixels in the grayscale as follows:(1)$$X_{av}=\frac{X_{crX}\times X_{crY}}{X_{g}}$$
where $X{\;}_{crX}$ indicates the pixel count in the x dimensions of $Y_{contextual}$, $X_{g}$ denotes the gray level count in the $Y_{contextual},$ and $X_{crY}$ shows the pixel count in the y dimensions of $Y_{contextual}.$
(2)$$X_{acis}=\frac{X\sum c}{X_{g}}$$

The distributed pixel is evaluated as follows:(3)$$Pd=\frac{X_{g}}{X_{lp}}$$

In Equation (3), $X_{lp}$ denotes the residual number of clipped image pixels.

### 3.2. Feature Extraction Using CapsNet

For the feature extraction process, the EOADL-PCDC technique uses the CapsNet model. The Conv layer, PrimaryCaps layer, and DigitCaps layer are the three fundamental layers of CapsNet [23]. Furthermore, using three FC layers, CapsNet forms a reconstructed stage. The Conv layer extracts the main key of input images. The original Sabour structure chooses the ReLU function, with 256 filters, or channels, having a size of 9×9 and stride of 1. Hence, the 256 channels given with the dimension of 20×20 from the Conv layers, the size of 9×9 with a depth of 256, and a stride of 2 are used, which leads to 32 PrimaryCaps layers of dimension 6×6×8, where PrimaryCap contains 8D (eight dimensions). This operation generates 1152 capsules. All the capsules have two different components: orientation and magnitude. The orientation comprises the instantiate properties or parameters of the entities, and the magnitude is the probability where the entity occurs.

When the capsule is calculated, it decides which data will be passed to the following layer. In addition, the squashing function attains a vector with 0 and 1 values, which ensures that a smaller vector takes values closer to 0 and a larger vector has a value below 1. Equation (4) demonstrates the squashing function, in which $v_{j}$ denotes the output vector of the *j*th capsules and $s_{j}$ is its overall input.
(4)$$v_{j}=squash\left(s_{j}\right)=\frac{\|s_{j}{\|}^{2}}{1+\|s_{j}{\|}^{2}}\frac{s_{j}}{\left\|s_{j}\right\|}$$

For the PrimaryCaps layer, the overall input to $s_{j}$ capsules is a weighted sum through each prediction vector ${\hat{u}}_{j|i}$ from the capsule by multiplying the output $u_{i}$ in the below layer by the weighted matrices $W_{ij}$, which implies ${\hat{u}}_{j|i}=W_{ij}u_{j}$ and $s_{j}=\Sigma_{i}{\hat{u}}_{j|i}$. This function exploits a weight transform matrix that encodes the spatial significance and other relationships amongst the features of the present one and the low-level capsule. If the estimated prediction vector has a higher value than the potential parent, then the coupling coefficient $\left(c_{ij}\right)$ value is adjusted to select the right connection path through the dynamic routing mechanism. The coupling coefficient between the *i*th capsule and the capsule in the top layer sums to 1 and is defined by the routing softmax function, where initial logits $b_{ij}$ are log-likelihoods that *i*th capsule must be coupled with the *j*th capsule:(5)$$c_{ij}=softmax\left(b_{i}\right)=\frac{exp\left(b_{ij}\right)}{\Sigma_{k}exp\left(b_{ik}\right)}$$

### 3.3. Parameter Tuning Using EOA

The EOA is used to adjust the hyperparameter based on the CapsNet model. EOA is a newly established metaheuristic optimization strategy which attempts to keep the balance between the exploitation and exploration phases [24]. Exploitation is used to locally search the space to obtain the best solution and increase the search quality, while exploration is used to globally search the space but prevents the local optimal solution. The EOA technique is derived from the dynamic mass balance of the control volume system. Similar to other metaheuristic algorithms, EOA begins with population initialization, which is generated based on the dimensions of the feature size and the number of particles:(6)$$V\frac{dP}{dt}=QP_{eq}-QP+G$$

The equation signifies the random initial population as follows:(7)$$p_{i}^{initial}=p_{min\;}+rand_{i}\left(p_{max\;}-p_{min}\right)$$
where $P_{min\;}$ and $P_{max}$ denote the minimal and maximal concentration of particles, correspondingly, $p_{i}^{initial}$ shows the initial concentration vectors of the *i*th particles, n refers to the number of particles in the population, and $rand_{i}$ belongs to $\left[0,\;1\right]$.

The equilibrium state is used to conclude the optimization technique as it is globally enhanced, and there is no knowledge of optimization at the beginning. Consider four different particles that remain the best throughout the entire optimization algorithm. The number of particles selected is random and dissimilar from that in other optimization techniques. The five chosen objects that help to create a vector called equilibrium pooling are discussed below.
(8)$$p_{eq\vec{.}pool}=-p_{eq\vec{.}\left(1\right),Peq\vec{.}\left(2\right)l,Peq\vec{.}\left(3\right),Peq\vec{.}\left(4\right),Peq\vec{.}\left(avg\right)}$$

Eventually, an exponential term (*E*) updates the concentration and balance between exploitation and exploration, and finally aims at achieving better optimization. ε denotes the random vector and ranges within $\left[0,\;1\right]$, and is given as follows:(9)$$\vec{E}=e^{\vec{\varepsilon }\left(t-t_{0}\right)}$$

In Equation (9), t differs with the difference in iteration (*i*), which can be signified as:(10)$$t=\left(1-\frac{i}{\;\max i}\right)\left(k_{2}\cdot \frac{i}{\;\max i}\right)$$
where i shows the existing iteration and maxi denotes the maximum iteration number. $k_{2}$ denotes the parameter that manages the exploitation capability. The succeeding formula demonstrates that if the searching performance is delayed by augmenting the exploitation and exploration capabilities, then the convergence is easier to obtain.
(11)$$t_{0}=\frac{1}{\in }ln\left(-k_{1}sign\left(m-0.5\right)\left[1-e^{-\in .t}\right]\right)+t$$

In the above equation, $k_{1}$ shows the exploration capability. The greater the value of $k_{2}$, the lower the exploration and the higher the exploitation capability. $sign\left(m-0.5\right)$ denotes the direction of exploitation and exploration. The values of m lie within [0, 1]. The amended version of Equation (9) is given below:(12)$$\vec{E}=k.sign\left(m-0.5\right)\left[e^{-\in t}-1\right]$$

Then, the generation rate is the next important stage, which gives an accurate solution to the optimization process by ensuring a better exploitation stage. There exist several algorithms for computing the generation rate amongst well-known models for 1D space, as given below.
(13)$${\vec{H}}_{G}={\vec{H}}_{0}\cdot e^{\vec{\in }\left(t-t_{0}\right)}$$

In Equation (13), $H_{0}$ refers to the initial value and ε denotes the decay constant:(14)$${\vec{H}}_{G}={\vec{H}}_{0}\cdot \vec{E}$$
(15)$$E\vec{-}0=GCP\left(Peq-\in P\right)$$
(16)$$GCP=\left\{\begin{array}{ll} 0.5\cdot m & if\;m>GP\\ 0 & else \end{array}\right.$$

Now, GCP refers to the generation control parameter. Finally, the EOA updating rule can be given in the following equation:(17)$$P=p_{eq}+\left(P-p_{eq}\right)E+\frac{F}{\in \;V\left(1-E\right)}$$

The EOA system derives an FF to accomplish higher efficacy of classification. It determines a positive integer to be the greater outcome of the solution candidate. Thus, the decay of the classifier error rate is assumed as an FF as follows:(18)$$fitness\left(x_{i}\right)=ClassifierErrorRate\left(x_{i}\right)\;\;\;\;\;\;\;\;\;=\frac{no.\;of\;misclassified\;samples}{Total\;no.\;of\;samples}*100$$

### 3.4. Image Classification Using SBiLSTM Model

The SBiLSTM model is applied for prostate cancer classification. BiLSTM is obtained via the data sequence of time-dependent input, together with latent relations between the input features and the destination [25]. It is proposed to record the long-term prior experience and manage it by applying memory units. LSTM provides several benefits while analyzing and projecting time series data. In the network module, a chain structure exists in LSTM and RNN. The RNN model encompasses a single neuron structure, while the LSTM model includes cells with three gates, namely, forget, output, and input gates. The succeeding equation demonstrates the computation method for the above three gates:(19)$$input\;\left(t\right)=\sigma \left(W_{i}x\left(t\right)+V_{i}h\left(t-1\right)+b_{i}\right)$$

In Equation (19), $W_{i}$ and $y_{i}$ signify the weights of the input gate, $h\left(t-1\right)$ symbolizes the output of prior cells, $x\left(t\right)$ signifies the input of existing cells, and σ denotes the sigmoid function:(20)$$forget\;\left(t\right)=\sigma \left(W_{f}x\left(t\right)+V_{f}h\left(t-1\right)+b_{f}\right)$$

This gate states that data in a cell must be forgotten, and $w_{f}$ and $y_{f}$ are the weights of the forgotten gate. By using the following expression, the update procedure was carried out:(21)$$\tilde{C}\left(t\right)=tanh\left(W_{c}x\left(t\right)+V_{c}h\left(t-1\right)+b_{c}\right)$$
(22)$$C\left(t\right)=forget\;\left(t\right)*C\left(t-1\right)+inpuf\left(t\right)*\tilde{C}\left(t\right)$$

Equation (21) shows the candidate unit for memory which generates present data. Moreover, Equation (22) illustrates the procedure of renewing conditions of the cell. $iVV_{c}$ and $y_{c}$ show the weight of the substitute and existing condition and ∗ shows the Hadamard product.
(23)$$output\;\left(t\right)=\sigma \left(W_{0}x\left(t\right)+V_{0}h\left(t-1\right)+b_{0}\right)$$
(24)$$h\left(t\right)=output\left(t\right)*tanh\left(C\left(t\right)\right)$$

Equations (23) and (24) analyze the output gate. Initially, the sigmoid layer is obtained via the cell states to be outputted. Through the tanh function, the cell state upgraded is processed, and the upgraded states are multiplied using output (t) to attain $h\left(t\right)$. $y_{0}$ shows the weight of the output gate. These frameworks create a BLSTM network to extract data features. In comparison to classical LSTM, BiLSTM is used to extract more context data. Backwards and forward time series are used to gain data regarding the present timestamp in the prior period and the future to create accurate time series predictions. Figure 2 shows the structure of the BiLSTM model.

The BiLSTM layer is added in the SBiLSTM mechanism to the stacked layer. Accordingly, output representations from the stacking layer are transferred to FC and the regression layer determines the temperature values. The *ReLU* function is used in the output layer to mitigate gradient vanishing problems:(25)$$temperature=ReLU\left(W_{0}h_{f}+b_{0}\right)$$

In Equation (25),$\;W_{0}$ and $b_{0}$ correspondingly show the weighted matrices and bias in the regression layer. $h_{f}$ indicates the output of the FC layer:(26)$$Loss=\frac{1}{T}\overset{T}{\underset{r=1}{\sum }}{\left(\left|temperature_{r}-temperature_{r}^{\prime }\right|\right)}^{2}$$

The proposed model was trained through the BP model. 

## 4. Results and Discussion

The presented approach was simulated using the Python 3.6.5 tool. The prostate cancer classification outcomes of the EOADL-PCDC method were validated on the dataset [16], including 400 samples with two class labels as described in Table 2.

Figure 3 illustrates the confusion matrices made by the EOADL-PCDC method on 80:20 and 70:30 of the TR phase (TRAP)/TS phase (TESP). The experimental data specify the efficient identification of the Prostate and Brachytherapy samples under every class.

In Table 3 and Figure 4, the prostate cancer detection outcome values of the EOADL-PCDC method are shown at 80:20 of the TRAP/TESP. The stimulation values highlighted that the EOADL-PCDC system recognizes two classes. With 80% TRAP, the EOADL-PCDC technique gains average $accu_{y}$, $reca_{l}$, $spec_{y}$, $F_{score}$, and MCC of 99.69%, 99.70%, 99.70%, 99.69%, and 99.38%, correspondingly. Moreover, with a 20% TESP, the EOADL-PCDC method attains average $accu_{y}$, $reca_{l}$, $spec_{y}$, $F_{score}$, and MCC of 98.75%, 98.61%, 98.61%, 98.73%, and 97.50%, correspondingly.

As shown in Figure 5, TR and TS $accu_{y}$ curves are defined to calculate the performance of the EOADL-PCDC technique on 80:20 of TRAP/TESP. The TR and TS $accu_{y}$ curves show the outcomes of the EOADL-PCDC system over numerous epochs. The figure provides relevant details about the learning process and generalizability of the EOADL-PCDC technique. The TR and TS $accu_{y}$ curves are enriched with the improvement in epoch count. It can be noticeable that the EOADL-PCDC algorithm yields enhanced testing $accu_{y}$ and has the proficiency to identify the patterns in the TR and TS datasets.

The comprehensive TR and TS loss values of the EOADL-PCDC system on 80:20 of TRAP/TESP over epochs are shown in Figure 6. The TR loss states the model loss minimized over epochs. Initially, the loss value is minimized as the model adapts the weight to minimize the predictive error on the TR and TS datasets. The loss curve exhibits the level where the model fits the training dataset. The TR and TS loss reduced progressively and showed that the EOADL-PCDC method efficiently learns the patterns given in the TR and TS datasets. The EOADL-PCDC technique adapts the parameters for the reduction in the variance between the predictive and original training labels.

As shown in Figure 7, the PR analysis of the EOADL-PCDC algorithm on 80:20 of TRAP/TESP is confirmed by plotting precision against recall. The simulated outcomes confirm that the EOADL-PCDC method attains increased PR values in each class. The figure describes the model learned for recognizing diverse class labels. The EOADL-PCDC technique achieves better experimental outcomes in the detection of positive samples with reduced false positives.

The ROC examination presented by the EOADL-PCDC model on 80:20 of TRAP/TESP is demonstrated in Figure 8, which has the ability to discriminate the classes. The figure provides insights into the tradeoff between the TPR and FPR rates over dissimilar classification thresholds and differing counts of epochs. It represents the accurately forecasted outcomes of the EOADL-PCDC method on the classification of dissimilar class labels.

In Table 4 and Figure 9, the prostate cancer recognition-simulated investigation of the EOADL-PCDC model is demonstrated at 70:30 of the TRAP/TESP. The observation data highlighted that the EOADL-PCDC models identify two classes. With 70% TRAP, the EOADL-PCDC system attains average $accu_{y}$, $reca_{l}$, $spec_{y}$, $F_{score}$, and MCC of 99.29%, 99.29%, 99.29%, 99.29%, and 98.57%, respectively. Moreover, with a 30% TESP, the EOADL-PCDC method gains average $accu_{y}$, $reca_{l}$, $spec_{y}$, $F_{score}$, and MCC of 99.17%, 99.14%, 99.14%, 99.17%, and 98.34%, correspondingly.

As presented in Figure 10, TR and TS $accu_{y}$ curves are shown to calculate the performance of the EOADL-PCDC method on 70:30 of TRAP/TESP. The TR and TS $accu_{y}$ curves show the outcomes of the EOADL-PCDC algorithm over dissimilar epochs. The figure provides significant details about the learning task and generalizability of the EOADL-PCDC method. With the improvement in epoch count, the TR and TS $accu_{y}$ curves acquire superior outcomes. The EOADL-PCDC method achieves maximum testing accuracy, which has the proficiency to identify the patterns in the TR and TS datasets.

Figure 11 illustrates the detailed TR and TS loss values of the EOADL-PCDC model on 70:30 of TRAP/TESP over epochs. The TR loss shows the model loss minimized over epochs. The loss values were minimized as a model adjusted the weight for the reduction in the predictive error on the TR and TS datasets. The loss curves exhibit the extent to which the model fits the trained dataset. The TR and TS loss reduced progressively and showed that the EOADL-PCDC method efficiently learns the patterns given in the TR and TS dataset. Note that the EOADL-PCDC algorithm adapts the parameters to diminish the dissimilarity between the original and predictive training labels.

As shown in Figure 12, the PR examination of the EOADL-PCDC model on 70:30 of TRAP/TESP is represented by plotting precision against recall. The experimental data confirm that the EOADL-PCDC methodology enhanced PR values under each class. The figure demonstrates that the model learns to detect dissimilar class labels. The EOADL-PCDC system accomplishes higher experimental outcomes in the detection of positive samples with diminished false positives.

The ROC investigation provided by the EOADL-PCDC method at 70:30 of the TRAP/TESP is demonstrated in Figure 13, which has the capability to discriminate the classes. The figure provides valued insights into the tradeoff between the TPR and FPR rates over dissimilar classification thresholds and varying counts of epochs. It obtains the accurately forecasted outcomes of the EOADL-PCDC method on the classification of several class labels.

The comparison study of the EOADL-PCDC method with current models is provided in Table 5 and Figure 14 [16]. The results indicate that the DT model achieves ineffectual performance. Next, the NB, SVM-Gaussian, SVM-RBF, and GoogleNet models obtained slightly enhanced outcomes. Although the AOADLB-P2C model reaches near-optimal performance, the EOADL-PCDC technique obtains maximum performance with $accu_{y}$, $sens_{y}$, $spec_{y}$, and $F_{score}$ of 99.69%, 99.70%, 99.70%, and 99.69%, correspondingly. These outcomes ensured the superior outcome of the EOADL-PCDC method on the PC classification algorithm.

## 5. Conclusions

In this article, we introduced an automated prostate cancer diagnoses method using the EOADL-PCDC method on MRI images. The main goal of the EOADL-PCDC technique lies in the recognition and categorization of prostate cancer. The presented model involves four major stages, namely, image preprocessing, CapsNet feature extraction, EOA-based hyperparameter tuning, and SBiLSTM-based classification. Primarily, the EOADL-PCDC technique applies image preprocessing to improve the image quality. In addition, the EOADL-PCDC technique follows CapsNet for the feature extraction process. The EOA-based hyperparameter tuning is used to enhance the performance of CapsNet. The EOADL-PCDC technique makes use of the SBiLSTM model for prostate cancer classification. A comprehensive set of simulations of the EOADL-PCDC technique was tested on the benchmark MRI dataset. The experimental outcomes revealed the superior performance of the EOADL-PCDC method over the existing techniques under various metrics. In the future, the EOADL-PCDC method can be further optimized to accommodate larger and more diverse datasets, enhancing its generalizability across different patient populations. Additionally, exploring the model’s potential for real-time or near-real-time diagnosis in clinical settings could significantly contribute to improving patient outcomes through timely interventions.

## Figures and Tables

**Figure 1 biomedicines-11-03200-f001:**
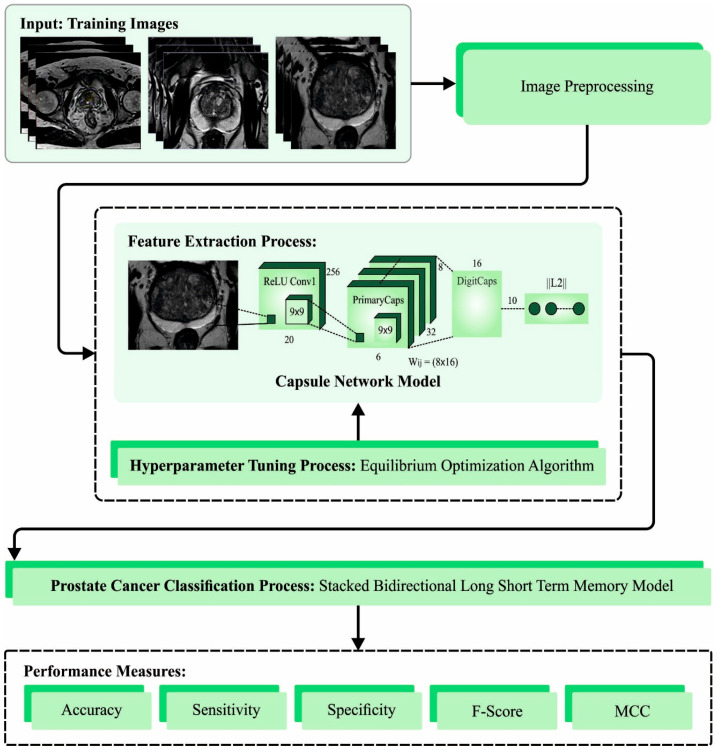
The overall working flow of the EOADL-PCDC technique.

**Figure 2 biomedicines-11-03200-f002:**
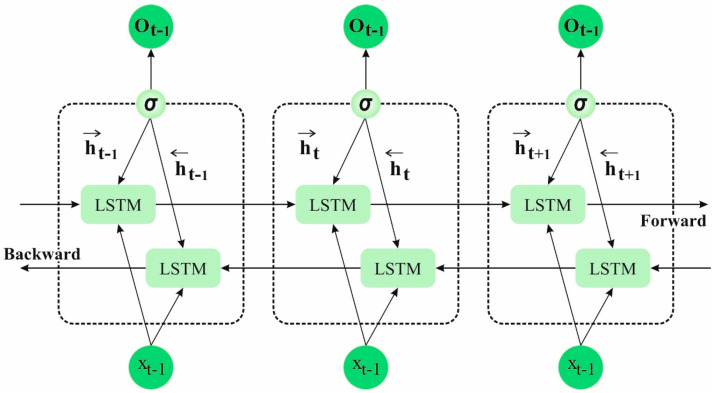
BiLSTM structure.

**Figure 3 biomedicines-11-03200-f003:**
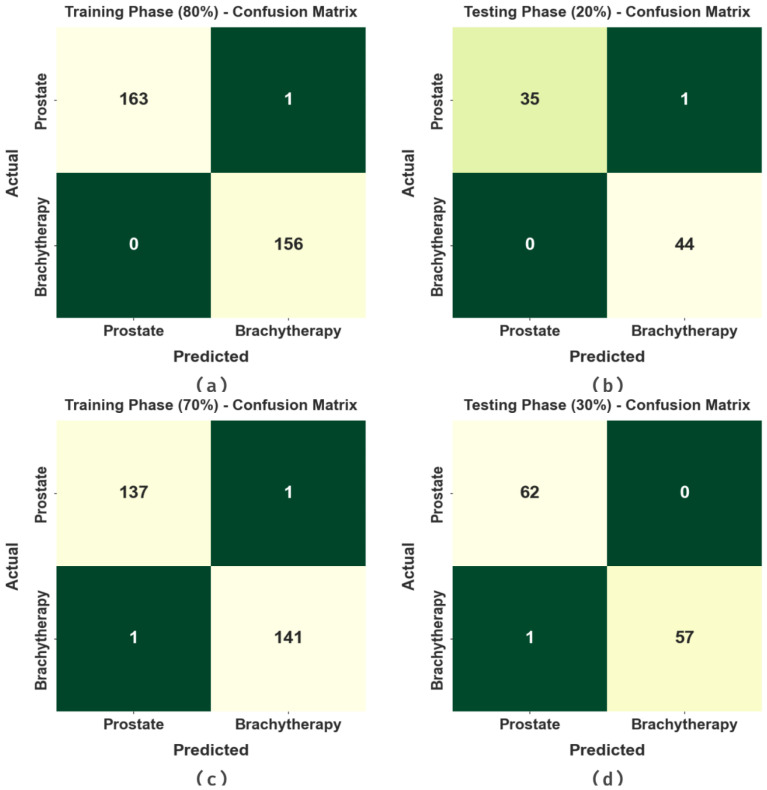
Confusion matrices of (**a**,**b**) 80% of TRAP/20% of TESP and (**c**,**d**) 70% of TRAP/30% of TESP.

**Figure 4 biomedicines-11-03200-f004:**
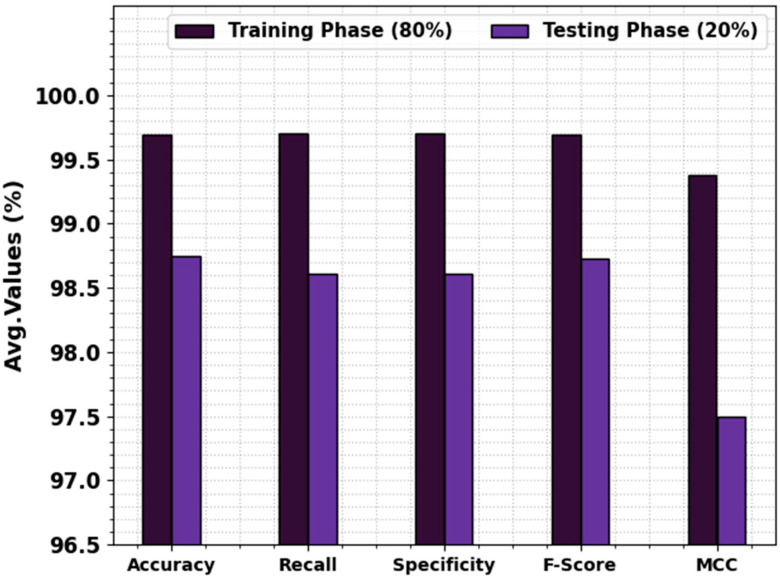
Average of the EOADL-PCDC system on 80:20 of TRAP/TESP.

**Figure 5 biomedicines-11-03200-f005:**
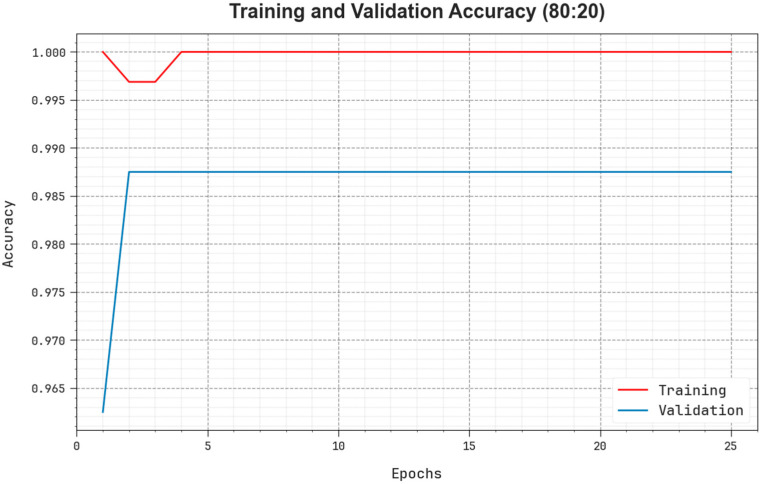
$Accu_{y}$ curve of the EOADL-PCDC algorithm on 80:20 of TRAP/TESP.

**Figure 6 biomedicines-11-03200-f006:**
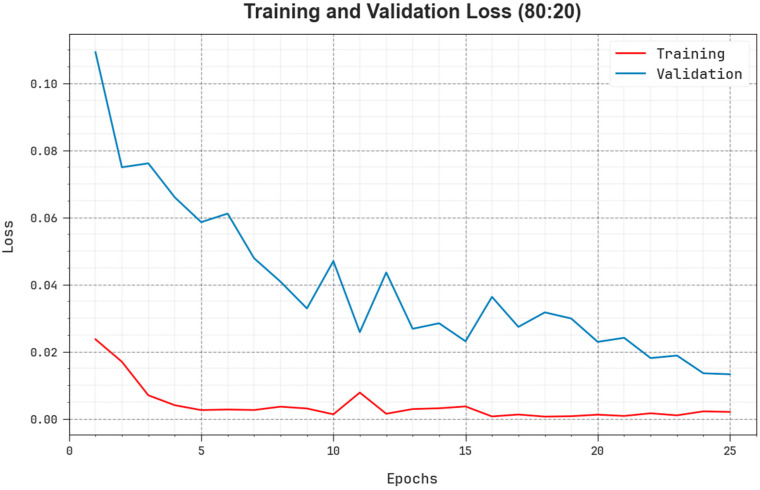
Loss curve of the EOADL-PCDC system on 80:20 of TRAP/TESP.

**Figure 7 biomedicines-11-03200-f007:**
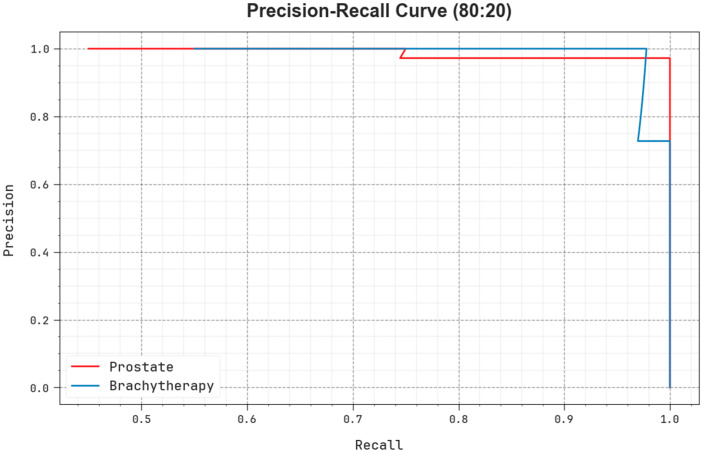
PR analysis of EOADL-PCDC technique on 80:20 of TRAP/TESP.

**Figure 8 biomedicines-11-03200-f008:**
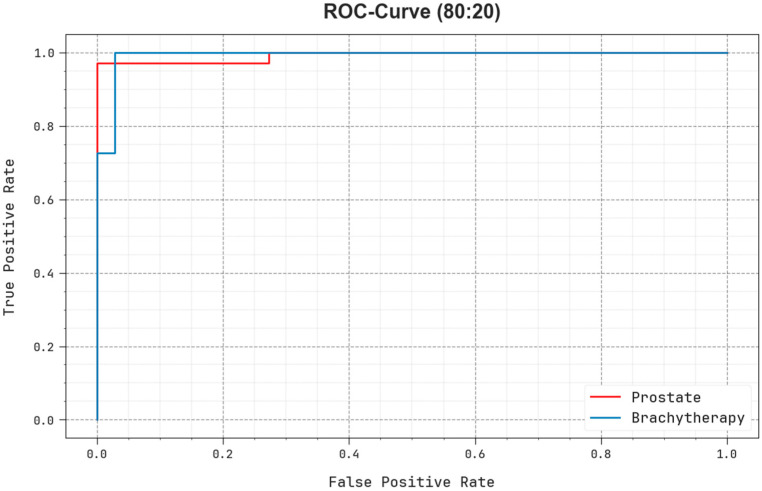
ROC analysis of the EOADL-PCDC technique on 80:20 of TRAP/TESP.

**Figure 9 biomedicines-11-03200-f009:**
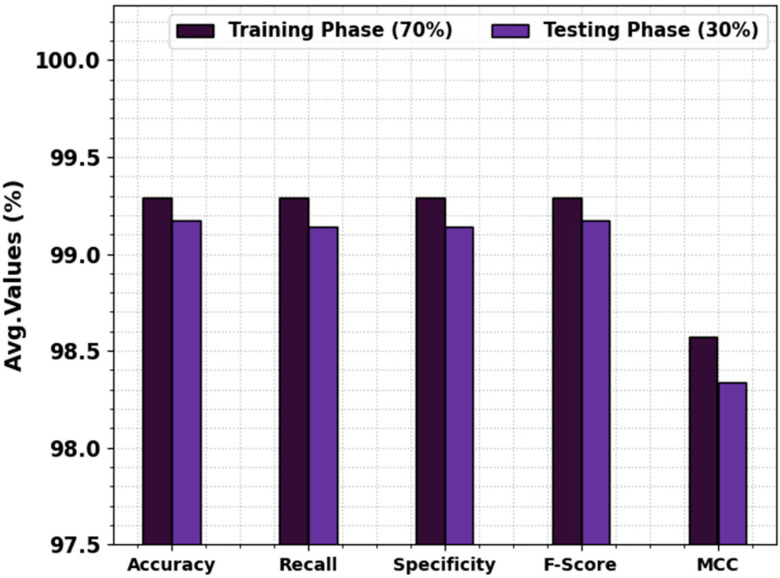
Average of the EOADL-PCDC method on 70:30 of TRAP/TESP.

**Figure 10 biomedicines-11-03200-f010:**
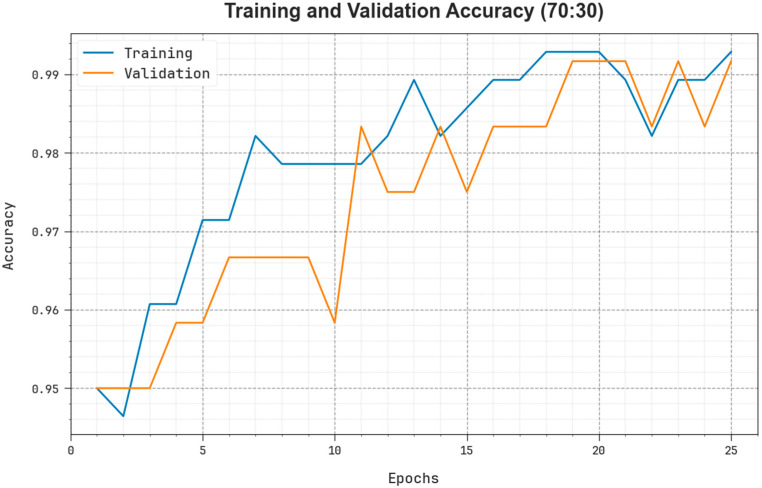
$Accu_{y}$ curve of the EOADL-PCDC technique on 70:30 of TRAP/TESP.

**Figure 11 biomedicines-11-03200-f011:**
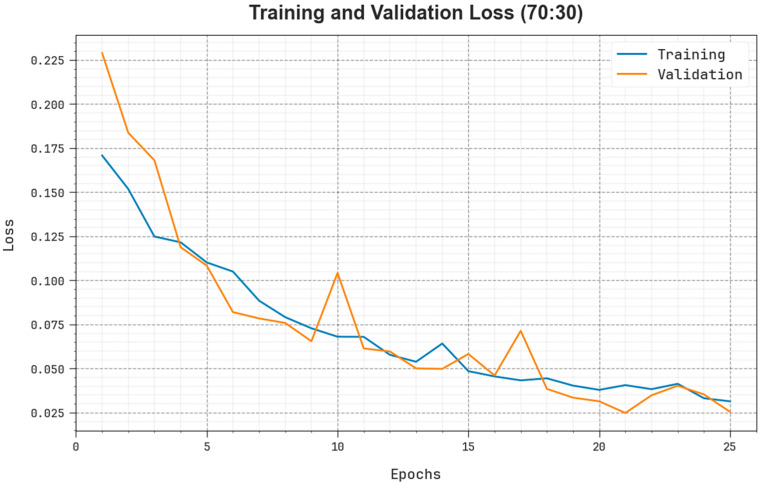
Loss curve of the EOADL-PCDC technique on 70:30 of TRAP/TESP.

**Figure 12 biomedicines-11-03200-f012:**
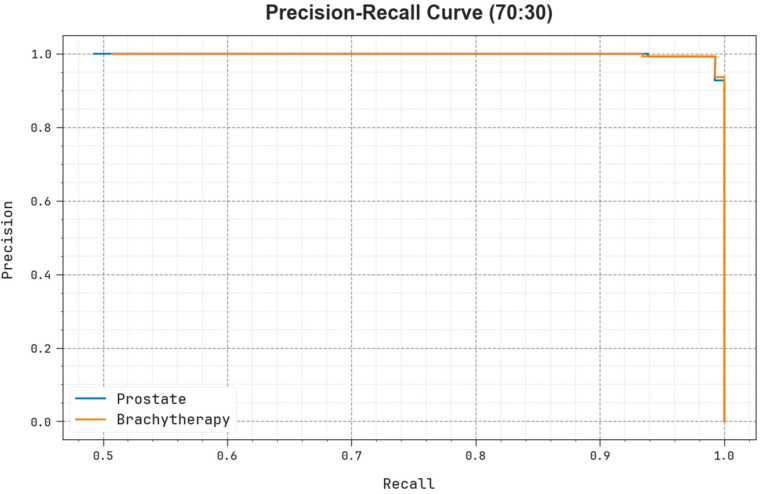
PR curve of the EOADL-PCDC algorithm on 70:30 of TRAP/TESP.

**Figure 13 biomedicines-11-03200-f013:**
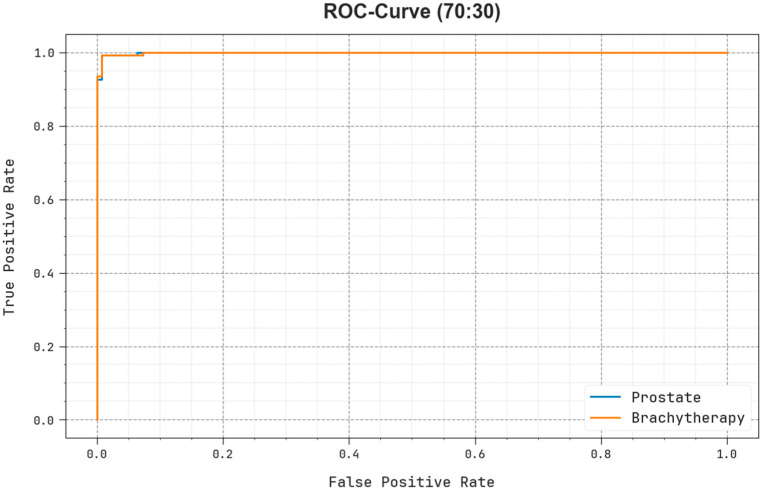
ROC curve of the EOADL-PCDC model on 70:30 of TRAP/TESP.

**Figure 14 biomedicines-11-03200-f014:**
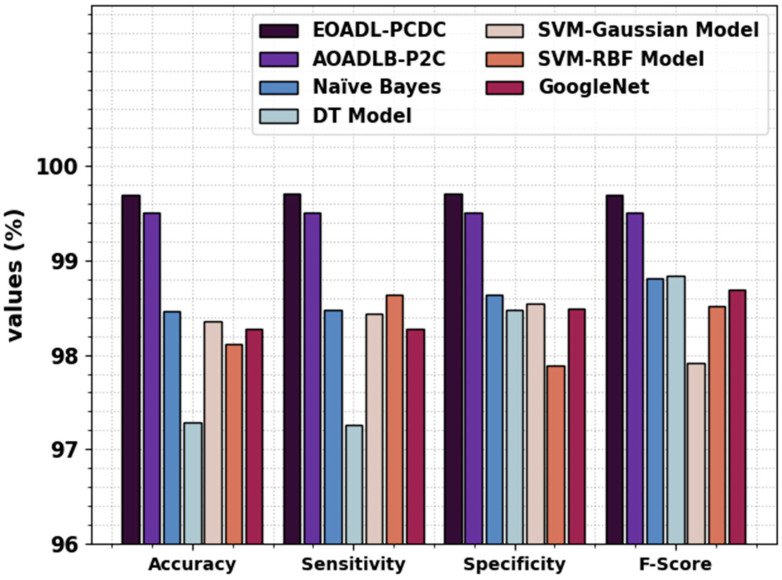
Comparative outcome of the EOADL-PCDC algorithm with recent systems.

**Table 1 biomedicines-11-03200-t001:** Summary of existing works.

Reference	Objective	Methodology	Dataset	Measures	Merits	Demerits
Singh et al. [11]	Classify PrC which belongs to the Gleason grade group	Faster RCNN with Inception-Resnet-V2	Prostate-2 dataset	Accuracy, sensitivity, specificity	Effective segmentation of lesion	Less experimentation
Yildirim et al. [12]	Aimed to detect the PI-RADS groups using mpMR images.	MobilenetV2, Efficientnetb0, and Darknet53	Own data	Accuracy	Can be used to reduce unnecessary biopsies	moderate to lower level agreement in PI-RADS scoring evaluation
Li et al. [13]	Classify PCa and prostate hyperplasia	Swin Transformer	Own data	AUC, ROC	Better predictive outcomes	Hard to detect the accurate area on MRI images
Lai et al. [14]	For auto-segmenting the prostate zone and cancer region	SegNet, DCNN	Own data	Accuracy, DSC, Recall	Superior results for PCa auto segmentation	Less amount of training data and requires model fine-tuning
Ye et al. [15]	Develop a prostate tumor diagnosis model	PSP-Net+VGG16	Own data	Accuracy	Superior in accuracy and processing time	Less experimentation
Ragab et al. [16]	Investigate MRI images for prostate cancer detection	DenseNet-161, LS-SVM, AOA	Own data	Accuracy, sensitivity, specificity, F-Score	Enhanced performance	Computational complexity analysis is needed
Moroianu et al. [17]	Identify PCa that spreads outside the prostate	U-Net	Own data	ROC, AUC	Parameter tuning is accomplished	Computational complexity analysis is needed
Bouslimi, and Echi [18]	Offer a fully automatic system for prostate detection	MultiResUnet	Radboudumc prostate cancer dataset	Accuracy	Enhanced performance	Less experimentation
Hassan et al. [19]	Detect prostate cancer using a fusion of different DL models	SVM, Adaboost, K-NN, and Random Forests	Own data	Accuracy	Examined by XAI	Requires fine-tuning of model parameters
Abbasi et al. [20]	Employ transfer learning model for prostate cancer detection	GoogleNet+ML classifiers	Harvard University prostate dataset	Sensitivity, specificity, PPV, NPV, and total accuracy	Enhanced performance	Requires fine-tuning of model parameters
Gavade et al. [21]	Classify prostate cancer	U-Net architecture+LSTM	I2CVB dataset	Accuracy, F1 score, precision, recall, ROC, dice	Reduce bias and enhance the generalization ability	Less experimentation

**Table 2 biomedicines-11-03200-t002:** Details of the database.

Classes	No. of Instances
Prostate	200
Brachytherapy	200
Total Samples	400

**Table 3 biomedicines-11-03200-t003:** Prostate cancer detection outcomes of the EOADL-PCDC algorithm on 80:20 of TRAP/TESP.

Class	$Accu_{y}$	$Reca_{l}$	$Spec_{y}$	$F_{score}$	MCC
TRAP (80%)					
Prostate	99.69	99.39	100.00	99.69	99.38
Brachytherapy	99.69	100.00	99.39	99.68	99.38
Average	99.69	99.70	99.70	99.69	99.38
TESP (20%)					
Prostate	98.75	97.22	100.00	98.59	97.50
Brachytherapy	98.75	100.00	97.22	98.88	97.50
Average	98.75	98.61	98.61	98.73	97.50

**Table 4 biomedicines-11-03200-t004:** Prostate cancer detection outcome of the EOADL-PCDC model at 70:30 of TRAP/TESP.

Class	$Accu_{y}$	$Reca_{l}$	$Spec_{y}$	$F_{score}$	MCC
TRAP (70%)					
Prostate	99.29	99.28	99.30	99.28	98.57
Brachytherapy	99.29	99.30	99.28	99.30	98.57
Average	99.29	99.29	99.29	99.29	98.57
TESP (30%)					
Prostate	99.17	100.00	98.28	99.20	98.34
Brachytherapy	99.17	98.28	100.00	99.13	98.34
Average	99.17	99.14	99.14	99.17	98.34

**Table 5 biomedicines-11-03200-t005:** Comparative outcome of the EOADL-PCDC method with existing techniques [16].

Methods	$Accu_{y}$	$sens_{y}$	$Spec_{y}$	$F_{score}$
EOADL-PCDC	99.69	99.70	99.70	99.69
AOADLB-P2C	99.50	99.50	99.50	99.50
NB	98.46	98.47	98.64	98.81
DT	97.29	97.26	98.47	98.83
SVM-Gaussian	98.36	98.43	98.54	97.91
SVM-RBF	98.12	98.63	97.89	98.52
GoogleNet	98.28	98.28	98.49	98.69

## Data Availability

Dataset information is available in the manuscript.

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
