# Peer review of "Equilibrium Optimization Algorithm with Deep Learning Enabled Prostate Cancer Detection on MRI Images"

_biomedicines, 2023, doi:10.3390/biomedicines11123200_

Round 1

Reviewer 1 Report (Previous Reviewer 2)

Comments and Suggestions for Authors

For the related work, a clearer, more structured representation of each work, possibly in tabular form, with comparative parameters such as technique used, dataset, and results obtained, would facilitate a quicker and more comprehensive understanding for the reader. Both advantages and limitation for different methods should be listed.

Grad-CAM or similar techniques are needed for model interpretation.

Comments on the Quality of English Language

There are some grammatical inconsistencies

Author Response

Dear reviewer, 
Thank you very much for reviewing our manuscript. 
We greatly appreciate your complimentary comments and suggestions. We believe all the comments have been accommodated in the revised draft. 
Please find attached a point-by-point response to your concerns. We hope that you find our responses satisfactory and that the manuscript is now acceptable for publication.

Sincerely yours, 
Eunmok Yang, PhD

Reviewer 2 Report (Previous Reviewer 3)

Comments and Suggestions for Authors

Thank you. No further remarks.

Author Response

Thank you for your time and efforts. 

Round 2

Reviewer 1 Report (Previous Reviewer 2)

Comments and Suggestions for Authors

The manuscript has been sufficiently improved to warrant publication in Biomedicines

This manuscript is a resubmission of an earlier submission. The following is a list of the peer review reports and author responses from that submission.

Round 1

Reviewer 1 Report

Comments and Suggestions for Authors

1. The study used MRI and ultrasound images for prostate cancer detection. It is crucial to assess the quality and size of the dataset used. Small or biased datasets can lead to overfitting, and the representativeness of the dataset is essential for the generalizability of the proposed method.

2. The proposed method explains why a certain decision is made based on input images. It's important to provide details on how this explainability is achieved and whether it can be trusted by medical professionals. Transparent and interpretable AI models are crucial in the medical field.

3. The study claims improved performance over existing methods, but it should provide a detailed description of the validation and evaluation process. Are there well-defined metrics used for comparison? What is the baseline performance, and how significant are the improvements?

4. It's essential to highlight that the performance of an AI model in a clinical setting may differ from simulation results. Has the proposed method been clinically validated, and if so, what were the outcomes? Clinical validation is a critical step in translating AI models to real-world healthcare applications.

5. The study should investigate the robustness of the proposed method to variations in data sources, acquisition protocols, and patient demographics. Models that perform well on one dataset may not generalize to others.

6. Deep learning models can be computationally intensive. It's essential to discuss the computational requirements of the proposed method, especially if it is to be used in a clinical setting.

7. While the study discusses improved performance, it should also highlight the potential clinical impact of the proposed method. Will it lead to earlier detection, better treatment decisions, or improved patient outcomes?

8. Deep learning is well-known and has been used in previous biomedical studies i.e., PMID: 37519050, PMID: 36642410. Therefore, the authors are suggested to refer to more works in this description to attract a broader readership.

9. To assess the true clinical utility of the model, it should be compared with the performance of human experts (radiologists or pathologists) in a real-world clinical setting.

10. Details about the architecture, hyperparameters, and training process should be provided to enable reproducibility of the results by other researchers in the field.

11. Uncertainties of models should be reported.

12. When comparing the predictive performance among methods/models, the authors should perform some statistical tests to see significant differences.

13. More discussions should be added, especially in terms of clinical insights of models.

14. How did the authors deal with overfitting problem of the model?

Comments on the Quality of English Language

English writing should be re-checked.

Reviewer 2 Report

Comments and Suggestions for Authors

The paper discusses a new method for detecting Prostate Cancer (PrC) using a combination of deep learning techniques applied to MRI and Ultrasound images. The method, named EOADL-PCDC, emphasizes not only the identification and classification of PrC but also elaborates on the reasons behind the decision-making process, given the input images.

1. Introduction:

Sentences are somewhat complex and could be simplified for better understanding.

Sentence starting with "Since, the practical concept of PrC..." (Line 37) is a bit convoluted and needs to be rephrased for clarity.

There are some grammatical inconsistencies that should be rectified, such as "can be the ability to resolve limits" (Line 41); a more appropriate phrasing would be "have the ability to address the limitations".

2. Related Works:

A clearer, more structured representation of each work, possibly in tabular form, with comparative parameters such as technique used, dataset, and results obtained, would facilitate a quicker and more comprehensive understanding for the reader.

It would be advantageous to explicitly mention whether the proposed model improves or addresses any of the gaps or limitations identified in the reviewed studies, thus highlighting the necessity and relevance of the proposed work.

While numerous studies and methodologies are mentioned, there’s a lack of explicit comparison between them and the presented approach. It would enhance understanding if the unique features or improvements of the proposed method over the reviewed works were explicitly highlighted.

4. Results:

While there is mention of a comparison with other models, a more comprehensive comparative analysis, including insights into why and how EOADL-PCDC outperforms the others, would enhance the rigor and depth of the evaluation.

Grad-CAM or similar techniques are needed for model interpretation.

If even the NB and DT model can achieve around 98% accuracy, what’s the point for your advanced model

Comments on the Quality of English Language

There are some grammatical inconsistencies

Reviewer 3 Report

Comments and Suggestions for Authors

I have the following comments:

-Which is the database from which prostate cancer cases were obtained? A full description of how patients were enrolled and/or of the origin of the database and its main features is required.

-Data should be compared and analyzed using inferential statistical analysis methods to demonstrate that statistically significant differences (p<0.05 or less) were obtained for the best algorithm.

-A Discussion section where the findings are interpreted and explained (along with a systematic comparison with the relevant scientific literature and current practice) is lacking. Overall, the manuscript should be restructured following a more compact and standardized scheme (e.g., Introduction, Materials and Methods, Results, Discussion and Conclusions), and parts of the current "Related works" section should be moved to either the updated Introduction or the Discussion sections.

-The English language of the manuscript should be extensively improved.

Comments on the Quality of English Language

An extensive editing of English language is required.

Reviewer 4 Report

Comments and Suggestions for Authors

The paper proposes a method for prostate cancer detection in MR images based on deep learning approach. There are many papers in this topic. Thus, Authors should show the novelty of the proposed method and contribution to the field.

1. The literature review lacks conclusion. It was not shown what gap in the current knowledge will be filled thanks to the conducted research, what scientific problem will be solved, how physicians will benefit thank to the proposed image analysis method.

2. The Authors assumed network validation scheme where data is split into training and test sets. For unknown reason, the results were shown for two different proportions of such splitting. Regardless of the lack of justification for such analyses, in this case 5-fold crossvalidation should have been used (the number of data is small - a total of 400 cases). I recommend repeating the experiments for such a network validation scheme.

3. Obtained results should be compared and discussed with similar studies already described in the literature.